# Non-communicable diseases risk factors among the forcefully displaced Rohingya population in Bangladesh

**Ayesha Rahman**[1], **Jheelam Biswas**[2]*, **Palash Chandra Banik**[2]

**1** Department of Public Health, American International University Bangladesh (AIUB), Dhaka, Bangladesh,
**2** Department of Non-communicable Diseases, Bangladesh University of Health Sciences (BUHS), Dhaka, Bangladesh

* jheelam.biswas@gmail.com

**Data Availability Statement:** All data relevant to the study are accessible in Mendeley data doi: 10.17632/mckh4gmgtn.1.

## Abstract

Rohingya refugees of Ukhiya, Cox's bazar are an unaccounted group of people who form the largest cluster of refugees worldwide. Non-communicable disease (NCD) alone causes 70% of worldwide deaths every year therefore, the trend of NCD among Rohingya refugees demands proper evaluation and attention. The objective of this study was to measure the NCD risk factors among a convenient sample of Rohingya refugees. This cross-sectional study was conducted among 249 Rohingya refugees living in Balukhali and Kutupalang Rohinga Camps at Ukhiya, Cox's bazaar using a survey dataset adapted from the WHO Stepwise approach to NCD Risk Factor Surveillance (STEPS). Data was collected through face-to-face interviews with a structured questionnaire. Anthropometric and biochemical measurements were done by trained medical assistants. Descriptive analysis was applied as appropriate for categorical variables. A Chi-square test and a student t test were performed to compare the categories. In general, the findings of NCD risk factors as per STEPS survey was 53.4% for tobacco use including smokeless tobacco, 2.8% for alcohol consumption, 23.7% for inadequate vegetable and fruit intake, 34.5% for taking extra salt, 89.6% for insufficient physical activity, 44.5% for confirmed hypertension, 16.9% for overweight, 1.2% for obesity and 0.8% for high blood sugar. Some modifiable non-communicable disease risk factors such as physical inactivity, tobacco smoking, extra salt with food, and hypertension are present among the Rohinga refugees in Bangladesh. These findings were timely and essential to support the formulation and implementation of NCD-related policies among the Rohingya refugees as a priority sub-population.

## Introduction

A vulnerable and socially disadvantaged group of people are considered to be more susceptible to non-communicable disease (NCD) risk factors [1, 2]. Over 15 million of all deaths worldwide are attributed to non-communicable diseases (NCD), which occur between 30 and 69 years of age, and almost a quarter of these untimely deaths are estimated to disproportionally occur in low and middle-income countries [3]. Tobacco use, physical inactivity, unhealthy diet

**Funding:** The authors of this study received no specific funding for this work.

**Competing interests:** The authors have no conflicts of interest to declare.

and harmful use of alcohol are considered to be the modifiable risk factors of NCDs that can be prevented by applying prior interventions [3, 4]. In recent times, global health policymakers are more concerned about the importance of the timely prevention, detection and correction of these modifiable risk factors to reduce the overall NCD mortality [5]. The World Health Organization has prioritized adequate monitoring and surveillance of the modifiable risk factors to overcome the NCD epidemics in low resource settings [6]. Refugees from different parts of the world are generally inflicted with poverty and social inequity [5]. Associations between poverty and social inequality with a high risk of morbidity and mortality from NCDs have been established in different studies [6–8]. One study shows that changes in nutrition and lifestyle behaviors contribute to type 2 diabetes mellitus among the migrant population [9]. Another study conducted on refugees from Iraq, Somalia, and Bhutan in the USA, found that limitation in receiving education is partly responsible for the higher prevalence of risk factors of diabetes among them than in the general population [10]. On the other hand, Burmese refugees in Australia have some level of awareness about the negative effects of smoking tobacco and chewing betel quid but little knowledge about the cessation of these habits [11]. Hypertension is widely prevalent among refugees and asylum seekers in Uganda, with a significant number of people unaware of their condition and consequently suffering from uncontrolled hypertension [12].

The Rohingya refugees are the world's largest group of stateless people seeking asylum in Bangladesh [13]. After August 2017, 6,93,000 adult refugees were forcefully migrated into Cox's bazar and was resettled forming new camps [14]. As of October 2019, an estimated 905,754 Rohingya refugees resided in Ukhiya, Cox's bazar in 34 camps [15]. A survey conducted by BRAC identified that food insecurity, inadequate access to health care are two major crises among the Rohingya refugees [16]. A US-based survey on re-settled Rohingya refugees from Myanmar shows a higher trend of chronic diseases like diabetes, hypertension and obesity along with widely prevalent risk factors of NCDs both in urban and camp settings [17]. Although there is a paucity of documentation of the prevalence of NCDs among the Rohingya refugees re-settled in Bangladesh, reviewing their unhealthy dietary pattern, physical inactivity resulting from a shift from rural to sedentary life, mental stress along with depression as a consequence of violently forced migration and some frequently observed NCD risk factors like smoking, using smokeless tobacco indoor air pollution, presence of established chronic diseases among them are highly presumable [18].

However, the Bangladesh government's health services and various NGOs currently working with the Rohingya refugees focus mainly on infectious diseases and mass vaccination among them. Therefore, the identification and prevention of infectious diseases have received the highest concern from the public health specialists to the government sectors as prime healthcare strategies, resulting in an insubstantial number of studies conducted on chronic diseases [15]. In addition, the treatment approach for NCDs is costly, multiphasic, and time-consuming, and it may pose a significant burden on the economy of a developing country like Bangladesh [15].

The present study aims to explore the non-communicable disease (NCD) risk factors in a convenience sample of the Rohinga refugees. Our study aims to act as baseline data for the larger studies so that the policymakers can put more emphasis on this area.

## Methods

### Study design and setting

This cross-sectional study was conducted among the Rohinga refugees resettled in two camps Balukhali and Kutupalang, at Ukhiya, Cox's bazaar using a convenient sampling technique.

Data collection was conducted from January 2019 to June 2019 through face-to-face interviews using a structured questionnaire.

## Sample size and criteria

The Rohingya refugees aged over 18 years residing in the above mentioned camps and voluntarily agreed to participate in the study were selected as study participants. Pregnant women, critically ill patients, and people with physical or mental disabilities were excluded from the study. A study conducted among the slum dwellers in Dhaka shows that at least one NCD risk factors was present in 19.5% of participants [19]. Overall prevalence of NCD risk factors were: 36.0% (95% CI: 31.82–40.41). According to that study p = 19.5%, q = (100–19.5) = 80.5% at 95% CI, z = 1.96 & d = 5%. So, sample size n = $pqz^2/d^2$ = (19.5x80.5)x$(1.96)^{2}/5^2$ = 241.21. Using this prevalence value as reference our calculated sample size was 241. Due to the deliberate participation of an adequate number of adult refugees in the study, our final sample size was 249.

## Data collection procedure

An adapted (mostly for socio-demographic background) questionnaire for this study was developed using step-I, step-II and, steps III of WHO STEPS protocol based on the 2010 STEPS survey in Bangladesh [20]. Bangla version of the STEPS questionnaire was orally translated into the local Rohingya language by an interpreter during data collection. Although the Rohingya refugees originally live in Mayanmar, the language they speak is similar to Cox's bazar's local dialect which assisted the local medical assistants to conduct the interviews. The questionnaire was pre-tested in the field before actual survey. It was administered by the interviewers and no proxy interview was taken. Three medical assistants were recruited from Balukhali and Kutupalong health camps and trained for three days for collecting physical and biochemical measurements as well as conducting interviews. It took twenty minutes on average to conduct each interview along with the physical and biochemical measurements.

## Ascertainment of the key variables

**Socio-demographic and behavioral variables (Step I).** Only the core questions about demographic information (step I) with few omissions such as date of birth (due to lack of accuracy on the part of the refugees), marital status etc were used to simplify the questionnaire for the limited time setting. The behavioral components (step I) included the core questions on tobacco, alcohol, physical activity, fruit and vegetable intake and extra salt intake. Information on previous history and treatment of hypertension were also obtained. The average time spent on moderate and vigorous physical activity was transformed into minutes per week. Physical activity less than 150 minutes per week was considered low. A standard measuring cup was used to obtain information on serving sizes of fruits and vegetables in a week.

**Physical and biochemical variables (Step II and III).** In step II, the physical measurements i.e height, weight and blood pressure were measured. The biochemical measurement in step III only included random blood sugar measurement. Systolic and diastolic blood pressure was measured using a manual aneroid sphygmomanometer with an average-sized cuff at sitting and lying positions. The average of the two measurements was used for analysis. Height was recorded in centimeters, and weight was recorded in kilograms using a portable digital weighing scale. Random blood sugar was measured using a standard glucometer.

According to the American Diabetes Association (ADA) guideline, the reference values of random blood sugar between 4.4–7.8 mmol/dl was considered normal, 7.8–11.1 mmol/dl was considered pre-diabetes, and $\geq$ 11.1 mmol/dl was considered diabetes.

According to the Seventh Joint National Committee (JNC 7), systolic blood pressure <120 mmHg was considered as normal, 120–139 mmHg as pre-hypertension, 140–159 mmHg stage I hypertension, and ≥160 mmHg considered stage II hypertension. Diastolic blood pressure ≤80 mmHg was considered normal, 80–89 mmHg as pre-hypertension, 90–99 mmHg as stage I hypertension and ≥100 considered as stage II hypertension.

According to the WHO guideline 2020, Body Mass Index (BMI) was classified as ≤18.5 as underweight, 18.5–24.99 as normal, 25–29.9 as pre-obesity, and ≥30 as Obesity (class I).

## Statistical analysis

The data was at first entered in Microsoft Excel 2010, and after editing and logical checking; it was analyzed in SPSS version 22.0. Descriptive analysis like frequency, percentage, mean and standard deviation were done as appropriate for the categorical variables. Chi-square test and student t-test were performed to see the association among the categories setting the α level at 0.05. 95% confidence interval was done to see the distribution among the population.

## Ethical considerations

This cross-sectional study was conducted in accordance with the Declaration of Helsinki (as revised in 2013). Ethical clearance was taken from the Research Review Committee and the Ethical Review Committee of the American International University Bangladesh (AIUB). Due permission was taken from the respected authority before entering the restricted camp areas. Written informed consent was taken from each respondent. Blood collection and anthropometric measurement were done by three trained medical assistants with due permission from the respondents.

## Results

Almost an equal number of respondents from both sex (118 men and 131 women) participated in this study with a mean age of *48.6±11.8* years. A majority (77.8%) of the respondents were aged 40 or above. More than half (67.8%) of respondents were unemployed, but the percentage of unemployment was higher (80.1%) among women. The percentage of illiteracy was very high. Eight (83.5%) out of ten respondents were illiterate (**Table 1**).

About one-fourth (22.9%) of the respondents were habitual smokers of whom 31.4% had at least ten years' habit of smoking. Though smoking was more prevalent among men (33.9%),

**Table 1. Socio-demographic characteristics of the respondents.**

| Variables | Men(n = 118) n(%) | 95% CI | Women (n = 131) n(%) | 95% CI | Both (n = 249) n(%) | 95% CI |
|---|---|---|---|---|---|---|
| **Age,** *years* | | | | | | |
| < 40 | 27 (22.7) | 15.1–30.3 | 30 (23.1) | 15.9–30.3 | 57 (22.4) | 17.2–27.6 |
| 40–50 | 36 (30.2) | 21.9–38.5 | 58 (44.7) | 36.2–53.2 | 94 (37.8) | 31.8–43.8 |
| >50 | 56 (47.1) | 38.1–56.1 | 42 (32.2) | 24.2–40.2 | 98 (40.0) | 33.9–46.1 |
| *Mean ± SD* | *50.6±12.4* | *48.4–52.9* | *46.8±11.1* | *45.0–49.0* | *48.6±11.8* | *47.2–50.2* |
| **Occupation** | | | | | | |
| Unemployed | 65 (54.7) | 45.7–63.7 | 104 (80.1) | 73.3–87.0 | 169 (67.8) | 62.0–73.6 |
| Employed | 54 (45.3) | 36.3–54.3 | 26 (19.9) | 13.1–26.7 | 42 (32.2) | 26.4–38.0 |
| **Educational status** | | | | | | |
| Illiterate | 86 (72.3) | 64.2–80.4 | 122 (93.8) | 89.7–97.9 | 208 (83.5) | 78.9–88.1 |
| Literate | 34 (27.7) | 19.6–35.8 | 8 (6.2) | 2.1–10.3 | 41 (16.4) | 11.8–21.0 |

than women (13%) more than half (59.5%) of the women were habituated with smokeless tobacco and betel leaf consumption. Alcohol consumption was not quite common among the participants. Three (34.5%) out of ten respondents consumed extra salt with their food, and the prevalence was almost the same in both genders. It was also noticeable that the majority (89.6%) of the respondents were sedentary workers. Most (76.3%) of them used to eat ≥5 servings of fruits and vegetables almost 6 days per week (**Table 2**).

About one-third (33.7%) of the participants claimed to have high blood pressure without having any obvious diagnosis, and only one-third (12.3%) take anti-hypertensive medication. The recorded average systolic and diastolic blood pressure was 115.4 ±15.4 mmHg and 74.88± 9.49 mmHg respectively. More than half of the respondents were found to have pre-hypertension according to their systolic (55.8%) and diastolic (68.3%) blood pressure. Three out of ten (32.2%) respondents were newly diagnosed as hypertensive.

The majority (74.7%) of study participants fell under the normal range of BMI (18.5–24.99). In addition, about one-sixth (16.9%) of the participants were documented as overweight and 1.2% as obesity class I. (**Table 3**).

Although the majority (84.7%) of the study participants showed an average blood glucose level, yet a significant percentage (14.4%) were also discovered to be pre-diabetic (7.8–11.1 mmol/l) from the primary measurement of random blood glucose level (**Table 4**).

Among all the risk factors (behavioral, measured and biomedical),insufficient physical activity (89.6%), tobacco use including smokeless tobacco (53.4%) and hypertension (44.5%) were more prevalent among both sexes (**Fig 1**).

**Table 2. Behavioral risk factors of NCD among the respondents.**

| Variables | Men(n = 118) | 95% CI | Women (n = 131) | 95% CI | Both (n = 249) | 95% CI | *P value*[*] |
|---|---|---|---|---|---|---|---|
| | n (%) | | n (%) | | n (%) | | |
| **Tobacco smoking** | | | | | | | |
| Habitual Smoker | 40 (33.9) | 25.9–43.1 | 17 (13.0) | 6.7–17.9 | 57 (22.9) | 17.7–28.1 | 0.86 |
| Non-smoker | 78 (66.1) | 56.9–64.1 | 114(87.0) | 81.2–92.8 | 192 (77.1) | 71.9–82.3 | |
| **Years of tobacco smoking** | | | | | | | |
| 10 years of smoking habit | 17 (31.4) | 27.9–37.3 | 16 (12.2) | 4.2–11.8 | 53 (21.3) | 15.4.-28.7 | 0.13 |
| 20 years of smoking habit | 46 (38.9) | 26.6–30.1 | 17 (13.0) | 6.7.2–17.9 | 63(25.3) | 11.2–20.2 | |
| **Smokeless tobacco and betel leaf consumption** | | | | | | | |
| Betel leaf chewing | 69 (58.5) | 50.8–68.6 | 48 (36.6) | 28.0–44.4 | 117 (47.0) | 41.2–53.6 | 0.42 |
| Tobacco leaf chewing | 46 (39.0) | 23.5–30.1 | 30 (22.9) | 15.2–29.4 | 76 (30.5) | 24.8–36.2 | 0.33 |
| **Alcohol consumption** | 5 (4.2) | 0.6–7.8 | 2 (1.5) | 0.0–3.6 | 7 (2.8) | 0.8–4.8 | 0.78 |
| **Extra salt intake** | 39 (32.8) | 24.3–41.3 | 47 (36.2) | 28.0–44.4 | 86 (34.5) | 28.6–40.4 | 0.48 |
| **Physical activity** | | | | | | | |
| Insufficient physical activity | 111(94.9) | 90.9–98.9 | 114(87.0) | 81.2–92.8 | 226 (89.6) | 85.8–93.4 | 0.76 |
| Moderate physical activity** | 0 (0) | 0.0–0.0 | 17 (13.1) | 7.2–18.8 | 17 (6.8) | 3.7–9.9 | |
| Heavy physical activity | 6 (5.0) | 1.1–8.9 | 0 (0) | 0.0–0.0 | 6 (2.4) | 0.5–4.3 | |
| **Fruit and Vegetable Intake** | | | | | | | |
| **Intake of food and vegetables per week** | | | | | | | |
| ≥6 days | 92 (77.3) | 69.7–84.9 | 98 (75.4) | 68.0–82.8 | 190 (76.3) | 71.0–81.6 | 0.07 |
| <6 days | 27 (22.7) | 15.1–30.3 | 32 (24.6) | 17.2–31.9 | 59 (23.7) | 18.4–29.0 | |
| **Servings of fruit and vegetable intake** | | | | | | | |
| ≥5 servings per day | 92 (77.3) | 69.7–84.9 | 98 (75.4) | 68.0–82.8 | 190 (76.3) | 71.0–81.6 | 0.07 |
| <5 servings per day | 27 (22.7) | 15.1–30.3 | 32 (24.6) | 17.2–32.0 | 59 (23.7) | 18.4–29.0 | |

** WHO defined as 150 minutes of moderate-intensity activity per week or equivalent; *Chi-square test was done

**Table 3. Measured risk factors of the respondents (n = 249).**

| Variables | Men(n = 118) | 95% CI | Women (n = 131) | 95% CI | Both (n = 249) | 95% CI | P value |
|---|---|---|---|---|---|---|---|
| | n (%) | | n (%) | | n (%) | | |
| **History of hypertension** | | | | | | | |
| Reported hypertension | 42 (35.3) | 26.7–43.9 | 42 (32.3) | 16.1–30.5 | 84 (33.7) | 27.8–39.6 | 0.05*** |
| H/O taking antihypertensive medications | 42 (35.3) | 26.7–43.9 | 14 (10.8) | 5.5–16.1 | 31 (12.3) | 8.2–16.4 | 0.53*** |
| **Systolic blood pressure (mmHg)** | | | | | | | |
| <120 | 28 (23.5) | 15.8–31.2 | 35 (26.9) | 19.3–34.5 | 63 (25.3) | 19.9–30.7 | 0.24*** |
| 120–139 | 60 (50.4) | 41.4–59.4 | 79 (60.8) | 52.4–69.2 | 139 (55.8) | 49.6–62.0 | |
| ≥140* | 31 (26.0) | 18.1–33.9 | 16 (12.3) | 6.7–17.9 | 47 (18.9) | 14.0–23.8 | 0.23** |
| *Mean ± SD* | *116.6±16.8* | | *114.3±13.9* | | *115.4±15.4* | | |
| **Diastolic blood pressure (mmHg)** | | | | | | | |
| <80 | 22 (18.4) | 11.4–25.4 | 24 (18.4) | 11.8–25.0 | 46 (18.4) | 13.6–23.2 | |
| 80–89 | 77 (64.7) | 56.1–73.3 | 93 (71.5) | 63.8–79.2 | 170 (68.3) | 62.5–74.1 | 0.06*** |
| ≥90* | 20 (16.8) | 10.1–23.5 | 13 (10.0) | 4.9–15.1 | 33 (13.3) | 9.1–17.5 | |
| *Mean ± SD* | *75.5±10.1* | | *74.2±8.9* | | *74.8±9.4* | | *0.64** * |
| Newly diagnosed# Hypertension | 51 (42.9) | 27.3–33.8 | 29 (22.3) | 17.7–28.5 | 80 (32.2) | 23.9–44.6 | 0.43*** |
| Confirmed Hypertension## | 93 (78.2) | 64.4–93.2 | 43 (33.1) | 28.9–44.3 | 111 (44.5) | 45.2–79.5 | 0.57*** |
| **Body Mass Index (BMI)** | | | | | | | |
| Underweight (<18.5 kg/m2) | 4 (3.4) | 0.1–6.7 | 14 (10.8) | 5.5–16.1 | 18 (7.3) | 4.1–10.5 | |
| Normal weight (18.5–24.99 kg/m2) | 93 (78.2) | 70.8–85.6 | 93 (71.5) | 63.8–79.2 | 186 (74.7) | 69.3–80.1 | 0.08*** |
| Overweight (25–29.9 kg/m2) | 20 (16.8) | 10.1–23.5 | 22 (16.9) | 10.5–23.2 | 42 (16.9) | 12.2–21.6 | |
| Obese (≥30 kg/m2) | 2 (1.7) | 0.0–4.0 | 1 (0.7) | 0.0–2.1 | 3 (1.2) | 0.0–2.7 | |
| *Mean ± SD* | *22.8±2.5* | *22.3–23.2* | *22.0±3.0* | *21.4–22.5* | *22.4±2.8* | *22.0–22.7* | *0.79** * |

* Stage I and II hypertension (systolic 140- ≥160 mmHg and diastolic 90- ≥100 mmHg) were merged due to insufficient number of participants in each groups

#includes those who have systolic and diastolic BP within hypertensive rages

## includes all newly diagnosed hypertensive cases and those who reported taking anti-hypertensive drugs previously

**Student t test was done

***Chi-square test was done

## Discussion

According to UNHCR 2017 report, the refugee population living in Ukhiya, Cox's bazar after the exodus of 2017, have an extremely vulnerable mental and physical state due to past experiences of trauma and persecution along with a bleak living situation in the overcrowded camps [18]. The present study indicates that the course of chronic diseases may rise considering the pattern of living and dietary condition of the camp dwellers at Ukhiya. The study has reported

**Table 4. Biochemical risk factors the respondents (n = 249).**

| Blood sugar (mmol/dL) | Men (n = 118) | 95% CI | Women (n = 131) | 95% CI | Both (n = 249) | 95% CI | P value |
|---|---|---|---|---|---|---|---|
| | n (%) | | n (%) | | n (%) | | |
| 4.4–7.8 | 101 (84.8) | 78.3–91.3 | 110 (84.6) | 78.4–90.8 | 211 (84.7) | 80.2–89.2 | |
| 7.8–11.1 | 18 (15.1) | 8.6–21.6 | 18 (13.8) | 7.9–19.7 | 36 (14.4) | 10.0–18.8 | 0.42* |
| >11.1 | 0 (0) | 0.0–0.0 | 2 (1.5) | 0.0–3.6 | 2 (0.8) | 0.0–1.9 | |
| *Mean ± SD* | *6.3±1.1* | *6.2–6.6* | *6.5±1.5* | *6.3–6.8* | *6.4±1.3* | *6.3–6.6* | |

*Chi-square test was done

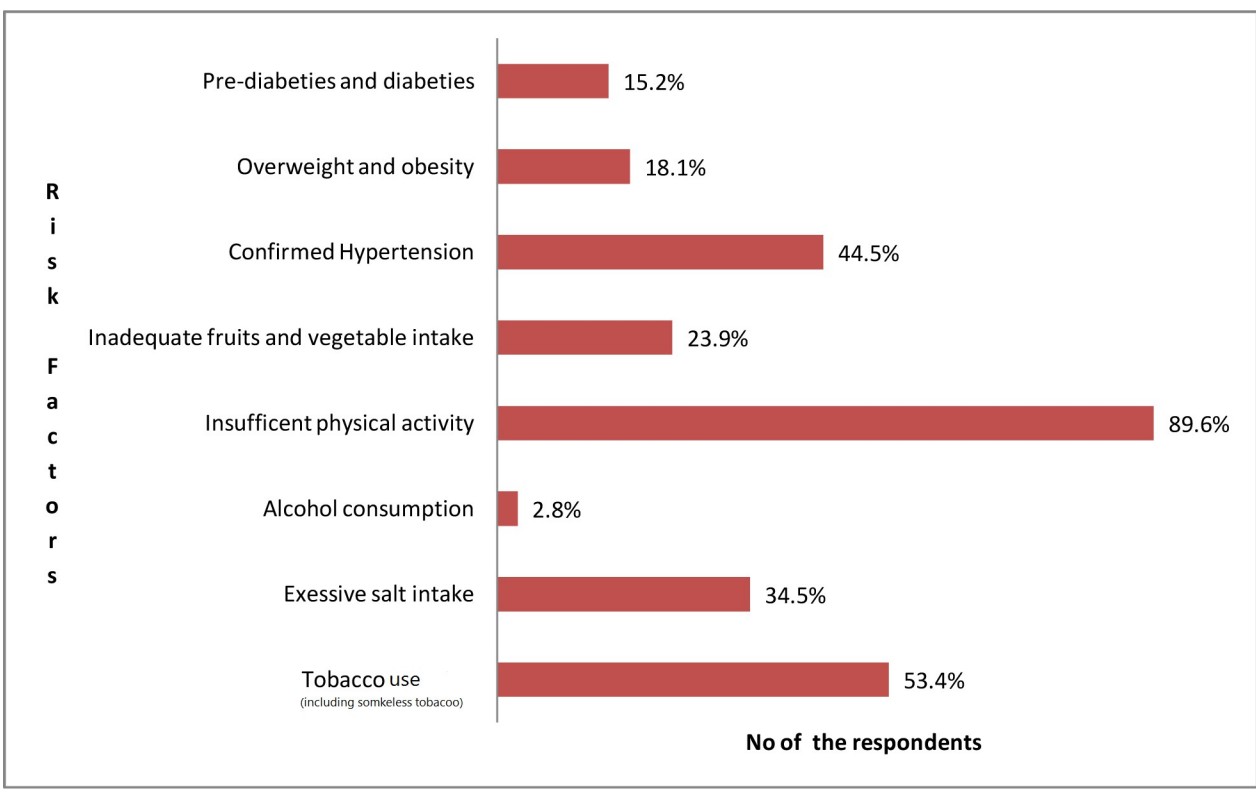

**Fig 1. Risk factors of NCD (n = 249).**

the behavioral risk factors of NCDs among the Rohingya refugees taking shelter in camp settings where most of the participants were unemployed (67.8%), and a significant percentage (83.5%) had no history of schooling. Non communicable diseases have been determined as a significant health challenge among many humanitarian set-ups around the world [21]. According to our study, the trend of NCDs among the Rohingya refugees is consistent with other refugee communities.

Screening of hypertension at refugee set-up has previously shown interesting outcomes in many places. One in five adults from Syrian refugees in Jordan was found to be hypertensive from self-documentation [22]. A need assessment conducted by BRAC on Rohingya refugees in 2018, reported that 51.5% of the refugees had hypertension and 14.2% had diabetes [15]. As per our study, one-third (32.2%) of the adult refugee participants were newly diagnosed as hypertensive and nearly half of them (44.5)%) were confirmed to have stage I and II hypertension. The percentage of confirmed hypertension was higher than the national survey on NCD risk factors among Bangladeshi citizens (7.9%) and the Syrian refugees in Northern Jordan (39.5%) [22, 23]. Only three out of ten hypertensive patients confirmed taking prescribed anti-hypertensive medicines regularly as opposed to their native Bangladeshi counterparts about half of whom take their anti-hypertensive medications regularly [23]. This percentage was also significantly lower than that in adult Syrian refugees (94.1%) who used to take medications on a regular basis [22]. Intake of extra salt with food is considered as the precipitating factor for hypertension, and more than one third of the respondents reported of consuming extra salt with their meals as a daily habit which was much higher than the Bangladeshi nationals (16.5%) [23]. The above scenario indicates that there is importance of screening for hypertension among the refugees in a community setup for early detection and treatment.

Although a negligible (0.8%) number of the respondents were diabetic according to the ADA reference values, 14.4% of the respondents were found to have a pre-diabetic level of random blood sugar. This finding is comparatively better than the Syrian refugees in Jordan, with 9.8% of respondents being reported as diabetic [22]. Since we could only measure random blood glucose, it could not appropriately measure the prevalence of diabetes among the participants. However, our finding reflects that the percentage of pre-diabetics is on the rise, which can create a burden of full-blown diabetes among the refugees in the near future.

Research has found that dependence on tobacco in any form has destructive health consequences [11]. Smoking was found among 22.9% of refugees in our current study which is close to the finding in native Bangladeshi people (23.5%) and nearly double in Palestinian refugees (36.6%) who are habitual smokers [23, 24]. In contrast, the picture is quite the opposite among female smokers. While only 0.8% of Bangladeshi women are smokers, around six out of ten Rohingya women have adopted smoking as a regular habit [23]. On the other hand, the percentage of consumption of smokeless tobacco is almost similar between the Rohingya refugees (30.5%) and the Bangladeshi nationals (32.0%) and higher than the Palestinian refugees (22.7%) in Syria [23, 24]. The refugee populations around the world are commonly seen to be a high-risk group for all form of tobacco addiction [24]. For instance, Palestinian refugees in Syria consume more cigarette and water pipe than the non-refugee residents in Lebanon [24]. In addition, the trend of alcohol and other substance use is high among displaced refugee communities around the globe [25]. For example, the prevalence of harmful alcohol consumption was around 36% among refugee men in Thailand and 23% among male Bhutanese refugees in Nepal. Similar findings have been found among internally displaced men in Uganda and Georgia with a prevalence of hazardous or harmful alcohol consumption of 32% [26]. However, we predicted that religious and cultural preferences have significantly lowered the alcohol consumption rate among the Rohingya refugees (2.8%), which is also compatible with the trend among the Bangladeshi nationals (1.3%). We could not extract information regarding other substance use due to communication barriers.

In the study, 16.9% of the participants were overweight having BMI between 25–29 kg/m$^2$, slightly lower than the Bangladeshi nationals (20.3%), and 1.2% of participants were obese with BMI above 30 kg/m$^2$ [23]. A US-based study on Somalian refugees has shown a thirty times more obese population than the present study on the Rohinga refugees [27]. As we have discussed earlier, the majority of Rohingya refugees are currently workless and living stagnantly, the number of inadequate physical activity (89.6%) in the study reflects the situation precisely. Nearly nine out of ten Rohingya refugees are physically inactive as opposed to three in ten Bangladeshi nationals (29.1%) who have insufficient physical activity [22]. It is presumed that the number of overweight/obesity may rise in the future considering their current condition of living. Moreover, large number of Rohingya refugees (76.3%) reported insufficient amount of fruit and vegetable intake which was a little less than the Bangladeshi nationals (89.6%) [23]. Physical inactivity, overweight and insufficient fruit and vegetable intake are risk factors for cardiovascular diseases, stroke, and cancer [23].

The research has several limitations which are worth mentioning. Primarily, we conducted a cross-sectional study to see the proportion of NCD risk factors among the refugees. It was conducted among the adult refugees of selected camp areas in Ukhiya, Cox's Bazar, which may not represent all the Rohingya refugees who have currently taken shelter in Bangladesh. The sampling method we used for the study is convenient sampling. We conducted the study among refugees from two pre-selected camps who voluntarily agreed to participate in the study. Since the law enforcement authority of government of Bangladesh maintains a strict surveillance system over the refugees we faced difficulty in conducting a broader study over them including every refugee settlement. In addition, we found many of the refugees who

experienced various trauma and persecution from the forceful migration not intent enough to participate in the interviews. This is why we preferred convenient sampling as a method which we understand not to be completely unbiased. Similarly, it was challenging to obtain confirmatory data for diabetes mellitus without measuring the fasting blood glucose level due to the absence of the participants' adequate cooperation. Hence, we had to rely only on random blood glucose measurement. In addition, single episode of blood pressure measurement may increase the chance of overestimation.

## Conclusion

This study depicts the status of NCD risk factors among a group of adult Rohingya refugees in Ukhiya, Cox's bazar. Some risk factors in particular, high blood pressure, smoking, consumption of extra salt with food, inadequate physical activity and insufficient fruit and vegetable intake are moderately high among the refugees. Presently, the Bangladesh government and international NGOs working with the refugees prioritize managing infectious diseases, which could be prevented by proper vaccination measures alone. But the course of NCDs among Rohingya refugees cannot be cured by one shot of injection. Even though the patient is cured, there is a chance of developing a disability consequently, which may become a potential threat of a sizeable economic burden for the government of Bangladesh [28] Therefore, the present study may shed some light on this aspect to encourage further research that will guide for implementation of the policies to curb NCD-related behavioral risk factors among the refugees.

## Supporting information

**S1 File. Ethical clearance letter.**
(PDF)

## Author Contributions

**Conceptualization:** Ayesha Rahman.

**Data curation:** Ayesha Rahman, Jheelam Biswas.

**Formal analysis:** Ayesha Rahman, Jheelam Biswas, Palash Chandra Banik.

**Funding acquisition:** Ayesha Rahman.

**Investigation:** Ayesha Rahman.

**Methodology:** Ayesha Rahman, Jheelam Biswas, Palash Chandra Banik.

**Project administration:** Ayesha Rahman.

**Resources:** Ayesha Rahman.

**Software:** Ayesha Rahman, Jheelam Biswas.

**Supervision:** Palash Chandra Banik.

**Validation:** Ayesha Rahman, Jheelam Biswas.

**Visualization:** Ayesha Rahman, Jheelam Biswas.

**Writing – original draft:** Ayesha Rahman, Jheelam Biswas, Palash Chandra Banik.

**Writing – review & editing:** Ayesha Rahman, Jheelam Biswas, Palash Chandra Banik.

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
