## [Decision Letter · Decision Letter 0]

5 Jul 2022

PGPH-D-22-00771

Non-communicable diseases risk factors among the forcefully displaced Rohingya population in Bangladesh

Dear Dr. Biswas,

Thank you for submitting your manuscript to PLOS Global Public Health. After careful consideration, we feel that it has merit but does not fully meet PLOS Global Public Health’s publication criteria as it currently stands. Therefore, we invite you to submit a revised version of the manuscript that addresses the points raised during the review process.

We look forward to receiving your revised manuscript.

Kind regards,

Biplab Datta, Ph.D.

Academic Editor

Journal Requirements:

Additional Editor Comments (if provided):

Please carefully address the comments and concerns raised by the reviewers. Additionally, justify and discuss why and how a convenience sample serves the purpose of adequately addressing the study aim. In the discussion section provide a comparison of the study findings with extant literature on refugee health and NCD risk factors among refugees.

Reviewers' comments:

Reviewer's Responses to Questions

**Comments to the Author**

1. Does this manuscript meet PLOS Global Public Health’s publication criteria? Is the manuscript technically sound, and do the data support the conclusions? The manuscript must describe methodologically and ethically rigorous research with conclusions that are appropriately drawn based on the data presented.

Reviewer #1: Partly

Reviewer #2: Yes

2. Has the statistical analysis been performed appropriately and rigorously?

Reviewer #1: Yes

Reviewer #2: Yes

3. Have the authors made all data underlying the findings in their manuscript fully available (please refer to the Data Availability Statement at the start of the manuscript PDF file)?

Reviewer #1: Yes

Reviewer #2: Yes

4. Is the manuscript presented in an intelligible fashion and written in standard English?

Reviewer #1: No

Reviewer #2: Yes

5. Review Comments to the Author

Reviewer #1: The authors of this study provided evidence on the status of non-communicable diseases (NCDs) risk factors in a population of forcefully displaced Rohingya population in Bangladesh using the World Health Organization STEPS approach. The findings of this study could be beneficial since it had focused on a vulnerable population and possibly the importance and risk of NCDs would be neglected and missed among them. Although the study is well designed and drafted it could benefit from some suggestions, especially in the methods of study which has many flaws. My comments for improvement of this submission are as follows.

1. General: language and grammar edit are essential for this manuscript. For example, using the past tense in the methods and results section is an essential part of a scientific presentation in an article.

2. Abstract, results, lines 38-44: the prepared part of the results in the abstract should reflect the results of the steps of the STEPS survey appropriately to make this part more informative.

3. Introduction: a focus on modifiable risk factors of NCDs could be beneficial in this section.

4. Methods, lines 94-104: the sampling method used for this seems to be a limitation and major issue needing explanation. How did authors made sure about including an un-biased sample of participants in this study?

5. Methods, lines 105-110: since the authors used the Bangla version of the STEPS questionnaire for this study and no adapted questionnaire was developed and implemented for the refugee population, it is a major limitation of this study and authors may expand in this issue in the limitation of the study.

6. Methods, lines 111-115: a separate part is needed in the methods section entitled study variables and the part should explain all variables used in the three steps of study and with appropriate referencing. Also, lines 121-130 could be moved to this section. Also, referring the readers to the supplementary files could be helpful in this regard.

7. Methods: the WHO STEPS survey includes much more physical measurements and laboratory tests in steps two and three. However, these steps were much briefer in this study. This issue needs explanation by authors.

8. Methods, data analysis: the method of calculating the prevalence values and 95% confidence intervals should be provided in this part of the methods section.

9. Results, line 139: the term “gender” should be replaced with “sex” since these two terms are technically different and could not be used interchangeably.

10. Results: adding some figures for data presentation in some parts of this section is suggested.

11. Discussion, lines 228-234: this section may benefit from some documents and programs of the United Nations High Commissioner for Refugees (UNHCR) for Rohingya refugees in Bangladesh and other relocated countries. Also, expanding this section on the possible implications of the study findings could be beneficial.

12. Declarations, lines 252-253: the referred website for online data repository could not be found with the data used in this study.

13. References: the bibliography needs a major revision making the citations uniform using the journal instruction guides for authors.

Reviewer #2: Manuscript title: Non-communicable diseases risk factors among the forcefully displaced Rohingya population in Bangladesh

Comments

The authors aim to highlight an important overlooked issue in an underserved group through their study. They aim to investigate and describe the state of NCD risk factors among Rohingya refugees. The authors used a cross-sectional design where data was collected using a questionnaire adapted from the WHO STEPs approach conducted via face-to-face interviews. The study needs some methodological clarification that will strengthen the inference of their findings once addressed. I present these concerns and some suggestions below:

1. The authors describe that interviews were conducted via face-to-face interviews using a semi-structured questionnaire. The description including ‘semi-structured’ and ‘interviews’ implies that qualitative data may have been collected as well. If that is not the case, then the authors should consider rewording the related statements in the methods and abstract sections.

2. Additionally, the statement ‘Descriptive analysis was done as appropriate for categorical and quantitative variables’ in the methods section of the abstract, should be revised as categorical variables are a type of quantitative variable, and thus the ‘and’ suggests another meaning.

3. Authors should include details on how the convenient sampling was conducted in the two camps. Since results show almost equal participation from both genders, did the authors apply any efforts to ensure equitable representation of males and females?

4. Length of residence at the camp is an important factor in this context. Were data on the length of stay at the camp collected? A participant residing at the camp for 3-5 years has been exposed to the NCD-related risk factors for a longer time compared to a resident that has been living there for 6 months-1 year. That is, if the authors presume that living in this specific environment and the related experiences of being forcefully displaced, are the major contributors to the NCD risk factors.

5. Also, were respondents asked about having risk factors prior to the migration? If not, a proxy for this would be information on family history of the NCDs or the risk factors. Was the latter collected? Prior history of NCD or their risk factors can be informative.

6. The authors state that ‘the predicted sample size was 241’. The methods used to arrive at the predicted sample size based on the prevalence of NCD risk factors in the Bangladeshi slums, should be identified and somewhat described.

7. Use of an existing questionnaire, albeit from the WHO to collect data is a good approach. How was the questionnaire adapted from the WHO STEPs approach to use in this study? Were items omitted etc.? How long was the questionnaire and how long, on average, did it take to conduct the interview?

8. In the ‘Data analysis’ section, is the reference to ‘quantitative variables’ meant to be ‘continuous variables’? Since the authors also reported means and Std Dev., another analysis besides chi-square test must have been performed to get these results.

9. The authors provide a description and definitions of the variables i.e., for BP, BMI, and diabetes. For BP and diabetes, American guidelines are followed. The WHO guidelines (if available for these two measures) would be a more appropriate reference given the context of the study, as done for the BMI.

10. Since “Blood collection and anthropometric measurement was done by expert medical technologist”, were they, or the persons that conducted the data collection, trained on interviewing procedures? How many persons supported the data collection?

11. For the study limitation, the cross sectional nature of the study

12. Can more advanced analysis i.e., regression be conducted perhaps with ‘number of risk factors ‘as an outcome?

13. Small concerns: grammatical errors

6. PLOS authors have the option to publish the peer review history of their article (what does this mean?). If published, this will include your full peer review and any attached files.

**Do you want your identity to be public for this peer review?** For information about this choice, including consent withdrawal, please see our Privacy Policy.

Reviewer #1: **Yes: **Sina Azadnajafabad, MD, MPH

Reviewer #2: No

---

## [Editor Report · Decision Letter 1]

18 Aug 2022

PGPH-D-22-00771R1

Non-communicable diseases risk factors among the forcefully displaced Rohingya population in Bangladesh

Dear Dr. Biswas,

Thank you for submitting your manuscript to PLOS Global Public Health. After careful consideration, we feel that it has merit but does not fully meet PLOS Global Public Health’s publication criteria as it currently stands. Therefore, we invite you to submit a revised version of the manuscript that addresses the points raised during the review process.

Most of the concerns raised during the first round of review were duly addressed in the revised version. However, there are some minor issues (listed below) that needs to be addressed before I can recommend this paper for publication.

We look forward to receiving your revised manuscript.

Kind regards,

Biplab Datta, Ph.D.

Academic Editor

Journal Requirements:

Additional Editor Comments (if provided):

1. Consider replacing the phrase “is likely to” with “may” in line 214.

2. Provide reference for the following statement: “Non communicable diseases have been determined as a significant health challenge among many humanitarian set-up around the world.”

3. Consider replacing the phrase “the native Bangladeshi population” with “their native Bangladeshi counterparts” in line 231.

4. Consider replacing the phrase “lower than the adult Syrian refugees” with “lower than that in adult Syrian refugees” in lines 232-233. Please revise similar grammatical issued throughout the manuscript.

5. Consider replacing “can be” with “is” in line 234.

6. Consider replacing “of taking” with “consuming” in line 235.

7. Replace “one (14.4%) out of ten” with “14.4%” in line 240.

8. Replace “percentage of diabetes” with “prevalence of diabetes” in line 243

9. Put the citation [21] after “While only 0.8% of Bangladeshi women are smokers” in line 250-251

10. Correct the typo “(32.0%)%)” in line 253 and “(1.3%))” in line 264

11. Since the study findings are not generalizable across different groups of Rohingya refugees, I recommend reframing the following sentence (lines 296-297), “This study depicts the status of NCD risk factors among the Rohingya refugees in Ukhiya, Cox’s bazar” as follows: “This study depicts the status of NCD risk factors among a group of adult Rohingya refugees in Ukhiya, Cox’s Bazar”.

12. Change the word “him” in line 303.

13. Consider replacing the word “will” with “may” in line 303 and in line 274

14. Give reference on how NCD management could be a sizeable economic burden for the government.

15. Regarding the convenience sample, change the statement “The present study aims to explore the non-communicable disease (NCD) risk factors among the Rohinga population,” in line 96-97 with “The present study aims to explore the non-communicable disease (NCD) risk factors in a convenience sample of the Rohinga refugees.” Consider making similar changes in the abstract as well.

16. please add more details than what is currently stated, “Using this prevalence value as reference our calculated sample size was 241.” In particular, provide information on the details of power analysis (e.g., significance level, statistical power, etc.).

17. It is unclear what is meant by “95% confidence interval was done to see the distribution among the population” in line 159-160 – please clarify/ rephrase.
---

## [Editor Report · Decision Letter 2]

7 Sep 2022

Non-communicable diseases risk factors among the forcefully displaced Rohingya population in Bangladesh

PGPH-D-22-00771R2

Dear Dr Biswas,

We are pleased to inform you that your manuscript 'Non-communicable diseases risk factors among the forcefully displaced Rohingya population in Bangladesh' has been provisionally accepted for publication in PLOS Global Public Health.

Best regards,

Biplab Datta, Ph.D.

Academic Editor